# Synaptotagmin 1 directs repetitive release by coupling vesicle exocytosis to the Rab3 cycle

Yunsheng Cheng[1,2†], Jiaming Wang[1,2†], Yu Wang[1,2], Mei Ding[1,3*]

[1]State Key Laboratory of Molecular Developmental Biology, Institute of Genetics and Developmental Biology, Chinese Academy of Sciences, Beijing, China; [2]University of Chinese Academy of Sciences, Beijing, China; [3]Center for Excellence in Brain Science, Chinese Academy of Sciences, Shanghai, China

**Abstract** In response to $Ca^{2+}$ influx, a synapse needs to release neurotransmitters quickly while immediately preparing for repeat firing. How this harmonization is achieved is not known. In this study, we found that the $Ca^{2+}$ sensor synaptotagmin 1 orchestrates the membrane association/disassociation cycle of Rab3, which functions in activity-dependent recruitment of synaptic vesicles. In the absence of $Ca^{2+}$, synaptotagmin 1 binds to Rab3 GTPase activating protein (GAP) and inhibits the GTP hydrolysis of Rab3 protein. Rab3 GAP resides on synaptic vesicles, and synaptotagmin 1 is essential for the synaptic localization of Rab3 GAP. In the presence of $Ca^{2+}$, synaptotagmin 1 releases Rab3 GAP and promotes membrane disassociation of Rab3. Without synaptotagmin 1, the tight coupling between vesicle exocytosis and Rab3 membrane disassociation is disrupted. We uncovered the long-sought molecular apparatus linking vesicle exocytosis to Rab3 cycling and we also revealed the important function of synaptotagmin 1 in repetitive synaptic vesicle release.

*For correspondence: mding@genetics.ac.cn

†These authors contributed equally to this work

Competing interests: The authors declare that no competing interests exist.

## Introduction

In nerve terminals, neurotransmitters are packaged into synaptic vesicles (SVs) and released by $Ca^{2+}$-induced exocytosis (*Sudhof, 2004*). Fast and precise neuronal reaction requires that SVs are clustered in front of the release site, the presynaptic active zone. SVs then dock at the active zone, where they are primed to adopt a competent "ready-for-fusion" state. An action potential induces the opening of $Ca^{2+}$ channels, and the rising $Ca^{2+}$ concentration stimulates SV-plasma membrane fusion. The basic membrane fusion reaction is mediated by evolutionarily conserved soluble NSF attachment protein receptors (SNAREs) and related proteins like Munc13 and Munc18 (*Weber et al., 1998*; *Sudhof, 2004*; *Brunger, 2005*; *Jahn and Scheller, 2006*; *Lang and Jahn, 2008*; *Jahn and Fasshauer, 2012*). However, the $Ca^{2+}$-sensing process that starts the SNARE engine is primarily carried out by the synaptotagmin family (*Chapman, 2002*; *Jahn and Fasshauer, 2012*). Through their C2 domains, synaptotagmins bind to $Ca^{2+}$, thus triggering membrane fusion (*Sudhof, 2004*). After exocytosis, SVs undergo endocytosis and recycling and are refilled with neurotransmitters for repeated rounds of release.

Rab3 protein is highly enriched in the nervous system and is specifically localized on SVs (*Fischer von Mollard et al., 1991*; *Fischer von Mollard et al., 1994*; *Geppert et al., 1994*; *Stahl et al., 1996*). Like other Rabs, Rab3 cycles on and off its target membranes according to its GTP- or GDP-bound state. On the vesicles, the active GTP-bound form of Rab3 is complexed with effector proteins like rabphilin and RIM (Rab3-interacting molecule) (*Shirataki et al., 1993*; *Li et al., 1994*; *Wang et al., 1997, 2000*), thus facilitating the recruitment/docking of SVs (*Nonet et al., 1997*; *Leenders et al., 2001*; *Tsuboi and Fukuda, 2006*). $Ca^{2+}$-induced exocytosis

**eLife digest** Neurons communicate with one another at junctions called synapses. The arrival of an electrical signal called an action potential causes calcium ions to enter the first cell, which in turn triggers the release of molecules called neurotransmitters into the gap between the neurons. The binding of these molecules to receptors on the second cell then enables the action potential to be regenerated.

For cells to respond rapidly and reliably to incoming electrical signals, they must maintain a supply of vesicles—the packages that contain neurotransmitters—close to the site where they are released from the first cell. The vesicles are held in contact with the cell membrane by a structure called the docking complex. A number of the proteins in this docking complex have been identified, including two that have been referred to as the 'yin and yang' of vesicle fusion: synaptotagmin, which promotes fusion, and Rab3, which limits it. However, little is known about how these and other proteins interact to keep vesicles docked at the membrane.

Cheng, Wang et al. have now clarified the docking process with the aid of experiments in nematode worms. In resting neurons that are not releasing neurotransmitters, synaptotagmin ('yin') binds to an enzyme called GAP and prevents it from converting GTP—an energy-storage molecule—into GDP. Given that Rab3 ('yang') requires a molecule of GTP to power its own activity, the actions of synaptotagmin ensure that Rab3 has enough energy to remain bound to other proteins within the docking complex.

However, when an action potential arrives, calcium ions enter the neuron, and some of them bind to synaptotagmin. This disrupts its interaction with the GAP enzyme, which thus becomes free to convert the GTP molecule bound to Rab3 into GDP. The loss of its energy source causes Rab3 to separate from its binding partners, and docking complex collapses. As a result, vesicles fuse with the membrane and release neurotransmitter molecules into the synapse.

Given that Rab3 and synaptotagmin have changed little over the course of evolution, it is highly likely that the same indirect interaction between these two proteins also regulates the release of transmitter at synapses in the mammalian brain.

can trigger disassociation of Rab3 from SV membranes through the GTP hydrolysis process (*Fischer von Mollard et al., 1991*; *Fischer von Mollard et al., 1994*; *Stahl et al., 1996*), but the underlying mechanisms are not clear. The GTP-to-GDP conversion not only removes Rab3 from SVs, but also simultaneously dissociates Rab3 from its binding effectors, which disassembles the docking complex so that both Rab3 and Rab3 effectors can be recycled for the next round of release (*Wang et al., 1997*, *2000*). Given their unique features, Rab3 and synaptotagmin have been considered as the Yin and Yang of membrane fusion, respectively (*Geppert and Südhof, 1998*). However, the functional regulatory interaction between synaptotagmin 1 and Rab3 cycling has not been identified nor has the mechanism by which this interaction is coupled to fast and repetitive neurotransmitter release.

Here, we found that synaptotagmin 1/SNT-1 in *C. elegans* is crucial for the SV association of RAB-3 protein. SNT-1 promotes the GTP-bound state of RAB-3 by inhibiting RAB-3 GAP. The catalytic subunit of RAB-3 GAP (RBG-1) localizes on SVs and directly binds to SNT-1. $Ca^{2+}$ treatment disrupts the direct association between SNT-1 and RBG-1. In addition, $Ca^{2+}$-binding activity of SNT-1 is essential for the dissociation of RAB-3 from SVs. Thus, our study reveals the pivotal dual role of synaptotagmin 1 in coupling SV exocytosis with the Rab3 membrane association and dissociation cycle.

## Results

### Search for components involved in RAB-3/SV association

In *C. elegans* motor neurons, RAB-3 fused with Green Fluorescent Protein (GFP) adopts a punctate pattern of localization along the length of the ventral and dorsal cords (*Mahoney et al., 2006*). This punctate RAB-3 pattern is similar to that of other SV proteins, including synaptobrevin and synaptotagmin (*Nonet et al., 1993*; *Nonet, 1999*; *Zhen and Jin, 1999*). A previous report also

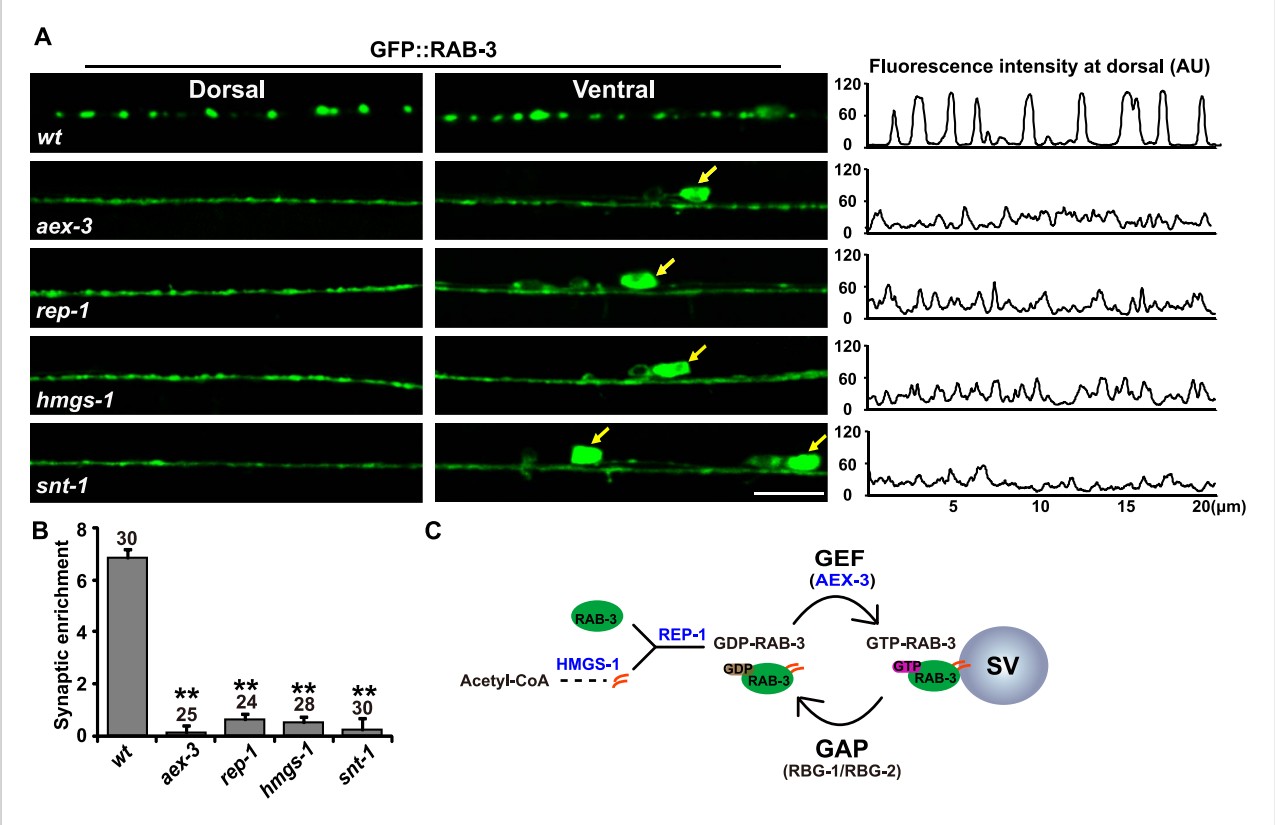

**Figure 1**. RAB-3 synaptic vesicle association requires SNT-1. (**A**) Punctate distribution of GFP::RAB-3 in *C. elegans* motor neurons in wild-type animals (top). The GFP::RAB-3 puncta become diffuse in *aex-3*, *rep-1*, *hmgs-1*, and *snt-1* mutants (lower panels). Yellow arrows indicate the cell bodies along the ventral cord. A representative line-scanning image for each genotype is shown in the right panel. (**B**) Quantification of the synaptic enrichment in wild-type, *aex-3*, *rep-1*, *hmgs-1*, and *snt-1* animals. Data are presented as mean ± SD; **p < 0.01. (**C**) Schematic representation of the RAB-3/SV association and dissociation cycle. Scale bar, 5 μm.

The following figure supplement is available for figure 1:

**Figure supplement 1**. *snt-1* is require for RAB-3 synaptic vesicle localization.

showed that most Rab3 protein is associated with SV membranes (*Fischer von Mollard et al., 1990*). In the absence of the RAB-3 GEF, AEX-3 (*Iwasaki et al., 1997*), GFP::RAB-3 no longer shows a punctate pattern and becomes diffusely distributed in neuron cell bodies and axons (*Figure 1A*). AEX-3 is responsible for converting RAB-3 protein from the membrane-dissociated GDP-bound form to the membrane-associated GTP-bound form (*Figure 1C*). Therefore, the punctate localization of RAB-3 in wild type likely represents the GTP-bound, SV membrane-associated form of RAB-3, while the diffuse GFP::RAB-3 signal may represent the dissociated GDP-RAB-3.

We speculated that mutations in components required for Rab3/SV association may lead to a diffuse RAB-3::GFP phenotype similar to that in *aex-3* animals. Hence, we conducted a genetic screen and isolated multiple mutants in which GFP::RAB-3 lost its punctate localization pattern. Through SNP mapping, complementation testing, and fosmid rescue, we cloned all of these mutations. Six of them (*xd58*, *xd137*, *xd142*, *xd143*, *xd148*, and *xd149*) turned out to be new alleles of *aex-3*. In addition, we obtained four *rep-1* alleles (*xd56*, *xd138*, *xd139*, and *xd140*) and three *hmgs-1* alleles (*xd128*, *xd129*, and *xd145*). *rep-1* encodes the sole Rab escort protein (Rep) (*Tanaka et al., 2008*). Rep proteins bind newly synthesized Rab proteins and facilitate the addition of geranylgeranyl groups to Rabs (*Seabra et al., 1992a, 1992b*; *Andres et al., 1993*) (*Figure 1C*). HMGS-1 is orthologous to the human hydroxymethylglutaryl-CoA synthase (HMGS) which is required for synthesis of the geranylgeranyl moiety (*Mehrabian et al., 1986*; *Shi and Ruvkun, 2012*) (*Figure 1C*). Thus, both *rep-1* and *hmgs-1* are critical for RAB-3 membrane targeting. The diffuse GFP::RAB-3 phenotype in

*rep-1* and *hmgs-1* mutants further suggested that the diffuse signal indeed comes from membrane-dissociated RAB-3 protein.

## Synaptotagmin/SNT-1 is required for localization of RAB-3 on SVs

In the meantime, we hypothesized that the molecules controlling the RAB-3 cycle may be associated with SV cycling. Thus, we systematically examined SV cycle-related mutants. Interestingly, we found that in *snt-1(md290)* animals, which lack *snt-1* function, the GFP::RAB-3 puncta disappeared and the GFP signal was diffusely distributed throughout the neuronal processes, similar to *aex-3*, *rep-1*, and *hmgs-1* mutants (*Figure 1A,B*). The strong phenotypic similarity between *snt-1* and other RAB-3/SV-association defective mutants suggests that SNT-1 plays an important role in RAB-3/SV localization.

*snt-1* encodes the synaptotagmin 1 homologue in *C. elegans* (*Nonet et al., 1993*). Neuronal synaptotagmins function as $Ca^{2+}$ sensors for synaptic exocytosis, but their role in Rab3 localization has not been revealed. To determine whether loss of *snt-1* function indeed leads to the diffuse RAB-3 phenotype, we examined other *snt-1* alleles. We found that the *n2665*, *md220*, *md125*, and *md172* alleles of *snt-1* all display a diffuse GFP::RAB-3 phenotype similar to *md290* (*Figure 1—figure supplement 1A*). Both the *snt-1* and *rab-3* genes are broadly expressed in the nervous system (*Nonet et al., 1993*, *1997*). To determine whether *snt-1* influences RAB-3 in all neurons, we examined RAB-3 localization with a pan-neuronal marker P*rab-3*::GFP::RAB-3. In wild-type animals, GFP::RAB-3 displays a punctate pattern, while in *snt-1* mutants, GFP::RAB-3 is completely diffuse in neuronal processes, including the nerve ring, ventral cord, and dorsal cord regions (*Figure 1—figure supplement 1B,C*). These data indicate that the effect of *snt-1* on RAB-3 localization is widely preserved in the nervous system. In addition, the diffuse GFP::RAB-3 phenotype was fully rescued when wild-type *snt-1* was introduced into mutant animals (*Figure 1—figure supplement 1D*), suggesting that SNT-1 is indeed essential for localization of RAB-3 on SVs.

## *snt-1* affects RAB-3 localization through an SV-independent mechanism

The diffuse RAB-3 phenotype in *snt-1* may be caused by failure of SV clustering at the synaptic terminal. Therefore, we examined the localization of another synaptic vesicle protein SNB-1. SNB-1 is the *C. elegans* synaptobrevin homologue (*Nonet, 1999*). In worm DD and VD motor neurons, SNB-1 is distributed evenly in punctate structures along neuronal processes, similar to RAB-3 (*Figure 2A*) (*Zhen and Jin, 1999*). In *snt-1* animals, some of the SNB-1 puncta are enlarged, but the punctate distribution of SNB-1 is not altered (*Figure 2A,B*). This observation is consistent with recent findings (*Yu et al., 2013*), suggesting that SV clustering is probably not affected by *snt-1*.

*unc-104* encodes the cytosolic kinesin responsible for synaptic vesicle trafficking from cell bodies to nerve terminals (*Hall and Hedgecock, 1991*). In the absence of UNC-104 kinesin, few SVs are transported to synaptic termini, while neuron cell bodies have a surfeit of SVs. To further address whether the diffuse RAB-3 phenotype is indeed caused by the dissociation of RAB-3 from SVs in *snt-1* mutants, we performed a serial mutant analysis utilizing both GFP::RAB-3 and SNB-1::GFP markers. Because the *snt-1;unc-104* double mutant animals are arrested during larval development, the synaptic phenotypes were examined in newly hatched L1 animals (larval stage 1). In L1 animals, among DD, VD, and AS motor neurons, only DDs are born (*Sulston, 1976*; *Sulston and Horvitz, 1977*), and they form pre-synapses along the ventral cord. Thus, in wild-type L1 animals labeled by P*unc-25*::SNB-1::GFP, the GFP signal could only be detected along the ventral cord (*Figure 2C,C″*) but not the dorsal cord (*Figure 2C′*). In *snt-1* mutants, the SNB-1::GFP distribution is indistinguishable from wild type (*Figure 2D,D′, and 2D″*). In *unc-104* single mutants, we found that SNB-1::GFP accumulated in cell bodies (*Figure 2E,E″*, white arrow) and little GFP signal could be detected outside of cell bodies or on the dorsal cord (*Figure 2E′*), which is consistent with the role of UNC-104 in SV transport. In *snt-1;unc-104* double mutants, the SNB-1::GFP signal accumulated in cell bodies (*Figure 2F,F″*, white arrow) like in *unc-104* single mutants, suggesting that further removal of SNT-1 in *unc-104* mutant animals does not alter the dependence of SVs on UNC-104 for intracellular trafficking. Interestingly, the effect of *snt-1* or *snt-1 unc-104* mutations on RAB-3 and SNB-1 is quite different. As shown in *Figure 2H,J*, GFP::RAB-3 is still diffuse in cell bodies and axons in both *snt-1* and *snt-1 unc-104* animals, and GFP signal could be detected even in the non-synaptic dorsal cord region (*Figure 2H′ and 2J′*). In contrast, in *unc-104* animals, the RAB-3 puncta are retained within cell bodies, just like SNB-1 is in *unc-104* mutants (*Figure 2I,I″*). These data strongly

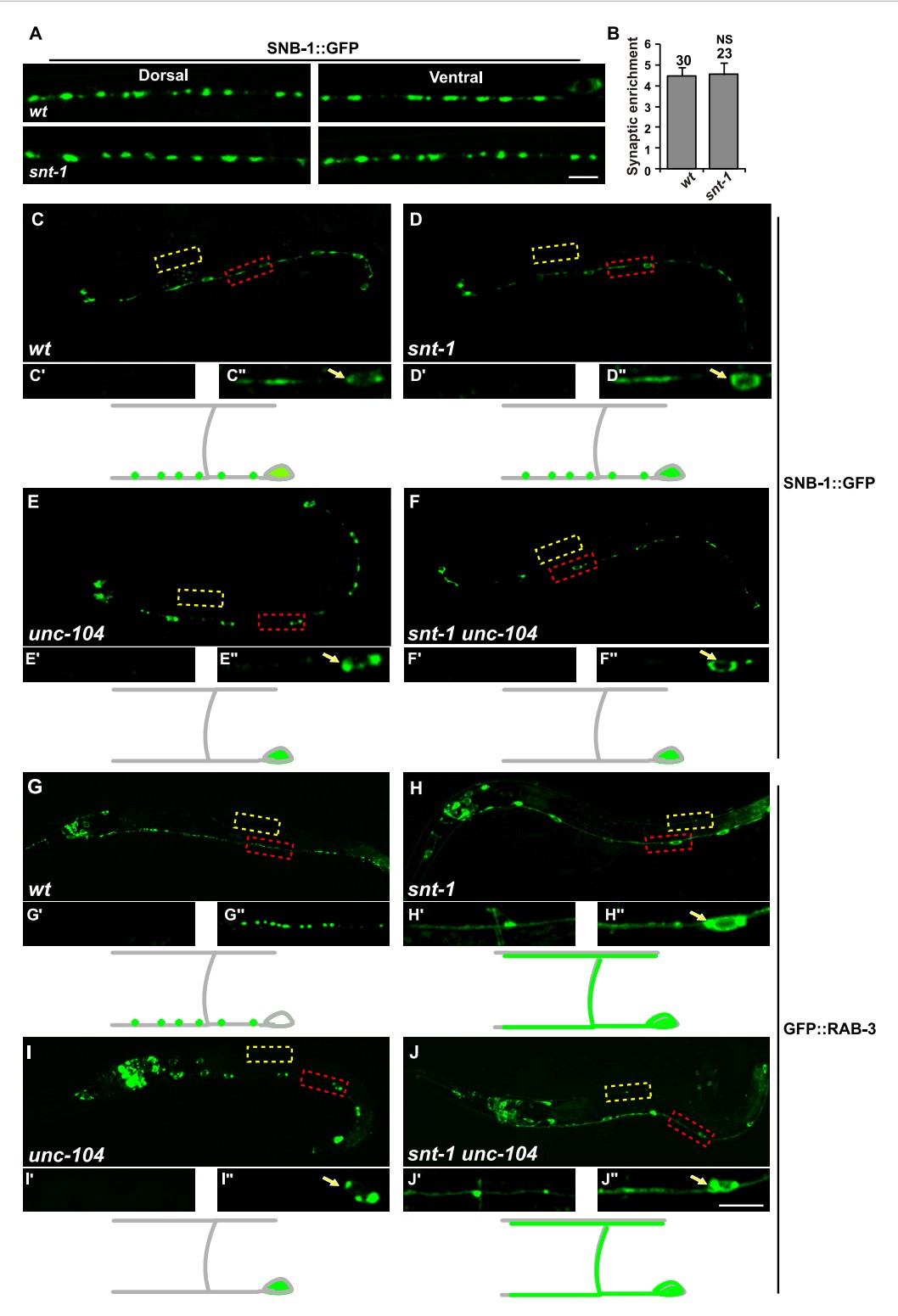

Figure 2. Synaptic vesicle clustering is unaffected by loss of *snt-1* function. (A) SNB-1::GFP puncta distribution in wild type and *snt-1* mutants. (B) The synaptic enrichment of SNB-1::GFP puncta is indistinguishable in wild type and *snt-1*. Data are presented as mean ± SD; NS, not significant. In both wild type (C, C' and C") and *snt-1* (D, D' and D"), the SNB-1::GFP puncta are present in the synaptic area on the ventral cord, which is outside of the cell body (C" and D"). In *unc-104* (E, E' and E") or *snt-1 unc-104* double mutants (F, F' and F"), SNB-1::GFP accumulates in the cell bodies on the ventral cord (E" and F"). (G) In wild type, GFP::RAB-3 is distributed in a punctate pattern in the

*Figure 2. Continued*

pre-synaptic regions on the ventral cord (**G″**). (**H**) GFP::RAB-3 is diffuse throughout the whole axon including both dorsal (**H′**) and ventral (**H″**) processes. (**I**) GFP::RAB-3 accumulates in ventral cell bodies (**I″**). (**J**) In *snt-1 unc-104* double mutants, GFP::RAB-3 is diffuse throughout the whole axon in both dorsal (**J′**) and ventral (**J″**) regions. Yellow boxes indicate part of the dorsal cord, which is enlarged in the lower left panels. Red boxes indicate part of the ventral cord, which is enlarged in the lower right panels (white arrows indicate DD cell bodies in the ventral cord). A schematic drawing of a DD neuron during the L1 stage is presented underneath the fluorescence images of each genotype, with the SNB-1::GFP or GFP::RAB-3 signal shown in green. Small green dots represent the pre-synaptic areas. Individual DD cell bodies are indicated as large ovals at the bottom right of each diagram. Scale bars, 5 μm.

support the notion that the diffuse phenotype of RAB-3 is not caused by the dispersion of SV clusters, but rather by the specific dissociation of RAB-3 from SV membranes.

## SNT-1 promotes the GTP-bound form of RAB-3

How does mutation of *snt-1* affect the SV membrane association of RAB-3? Previous studies showed that the localization of RAB-3 on SV membranes is tightly associated with its GTP-bound state (*Zerial and McBride, 2001*). Therefore, we tested whether the loss of RAB-3 from SVs in *snt-1* mutants is caused by reduction of GTP-bound RAB-3. The active GTP-Rab3 binds to the RBD domain of its effector RIM, while the inactive GDP-Rab3 does not. Previous reports demonstrated that the RBD domain of mammalian RIM2 could bind to the worm GTP-RAB-3 (*Wang et al., 1997*; *Mahoney et al., 2006*). Thus, we performed pull-down assays to examine the GTP-RAB-3 level in vivo. In wild-type worm lysates, the active GTP-bound form of RAB-3 protein was efficiently pulled down by GST-RBD (*Figure 3A*). In contrast, the amount of GTP-RAB-3 pulled down by RIM2 RBD was significantly reduced in *snt-1* lysates (*Figure 3A,B*). In the absence of RAB-3 GEF, the GDP-bound RAB-3 cannot be converted to the GTP-bound RAB-3. Indeed, in *aex-3* animals, the amount of RAB-3 that can be pulled down by GST-RBD is also greatly decreased (*Figure 3A,B*). Thus *snt-1*, similar to *aex-3*, affects the level of GTP-bound RAB-3 in vivo.

GTP-RAB-3 is associated with SV membranes, while GDP-RAB-3 is diffused in the cytosol. Thus, we performed cell fractionation experiments to further examine the GTP or GDP status of RAB-3. In wild type, RAB-3 is highly enriched in membrane fractions (*Figure 3C*), which is consistent with the SV localization of GTP-RAB-3. In contrast, the RAB-3 protein distribution is shifted to the soluble fraction when *snt-1* is removed, suggesting a cytosolic localization of RAB-3 in *snt-1* mutants (*Figure 3C*). These data suggest that *snt-1* indeed promotes the GTP-bound form of RAB-3.

## SNT-1 promotes the GTP-RAB-3 level by inhibiting GTP hydrolysis

How does loss of function of *snt-1* lead to the reduction of GTP RAB-3? One possibility is that SNT-1 may regulate the RAB-3 GTP-GDP cycle by promoting GEF activity. We performed the following experiments to test this possibility. Firstly, AEX-3 is the GEF molecule for both RAB-3 and RAB-27 (*Mahoney et al., 2006*). If *snt-1* indeed affects AEX-3 activity, we would expect that the localization of RAB-27 on SVs will be affected by the absence of SNT-1. We made a GFP::RAB-27 reporter and expressed it in motor neurons in worms. The GFP::RAB-27 protein is enriched in synaptic regions and displays a punctate expression pattern similar to RAB-3 (*Figure 3D*). In *aex-3* mutants, GFP::RAB-27 becomes diffuse, consistent with the role of AEX-3 as a GEF for RAB-27 (*Figure 3D*). In contrast, GFP::RAB-27 still displays a punctate distribution indistinguishable from wild type in *snt-1* mutants (*Figure 3D*), suggesting that the GEF activity of AEX-3, at least for RAB-27, is not altered by mutation of *snt-1*. Second, if SNT-1 promotes AEX-3 GEF activity, we would expect that increasing the *aex-3* expression level may rescue the diffuse RAB-3 phenotype in *snt-1* mutants. However, no such rescue was observed (*Figure 3E*). Lastly, we examined the expression level of AEX-3 and found that it was indistinguishable in wild-type and *snt-1* animals (*Figure 3F*). Together, these results suggest that it is unlikely that *snt-1* regulates the GTP-RAB-3 level by promoting RAB-3 GEF activity.

Alternatively, the decreased GTP-RAB-3 level may be caused by increased RAB-3 GTPase activity in *snt-1* mutants. Rab3 GTPase activity is greatly facilitated by Rab3-specific GTPase-activating protein (GAP). Rab3 GAP is composed of the catalytic subunit Rab3GAP1 and the noncatalytic subunit Rab3GAP2. *rbg-1* and *rbg-2* encode Rab3GAP1 and Rab3GAP2, respectively in worms

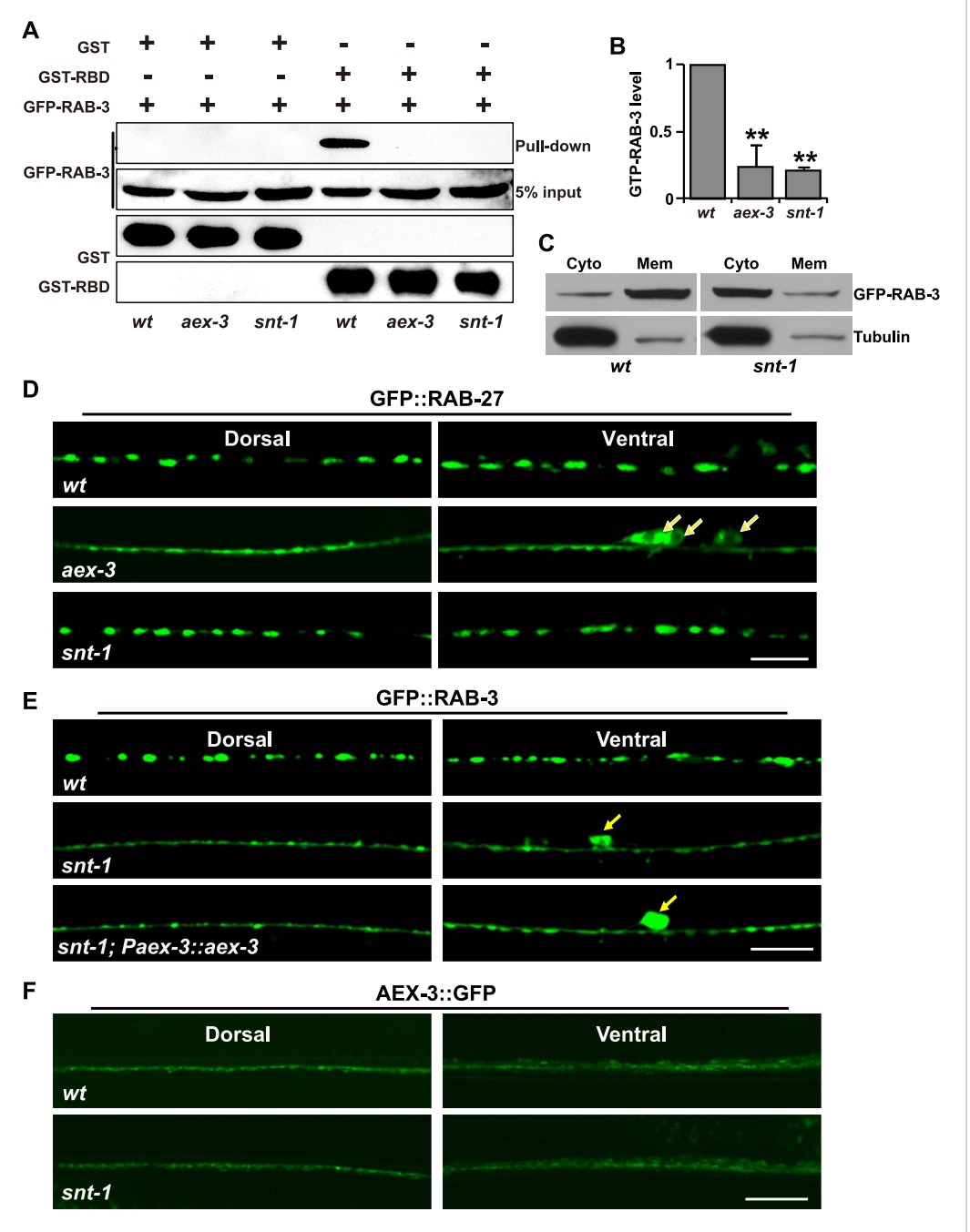

**Figure 3**. The GTP-bound form of RAB-3 is decreased in *snt-1* mutants. (**A**) A GST-fused RBD domain of RIM2 binds active GTP-RAB-3. The amount of GTP-RAB-3 pulled down by RBD is decreased in both *aex-3* and *snt-1* animals. (**B**) Quantification of the GTP-RAB-3 level in wild type, *aex-3*, and *snt-1*. Data are presented as mean ± SD; **p < 0.01. (**C**) The amount of GFP-RAB-3 in the cytosolic fraction is increased in *snt-1* mutants. (**D**) Localization of GFP::RAB-27 puncta is affected by mutation of *aex-3*, but not by mutation of *snt-1*. The white arrows indicate the cell bodies. (**E**) Over-expression of *aex-3* does not rescue the *snt-1* mutant phenotype. Yellow arrows indicate the cell bodies. (**F**) The AEX-3::GFP level is unchanged in *snt-1* mutants compared to wild type. Scale bars, 5 μm.

(*Figure 4—figure supplement 1A,B*) (*Fukui et al., 1997*; *Nagano et al., 1998*). In the absence of Rab3 GAP, the RAB-3 synaptic enrichment is enhanced, which is consistent with the role of Rab3 GAP in assisting GTP hydrolysis (*Figure 4—figure supplement 1C,D*). If RAB-3 GTP hydrolysis

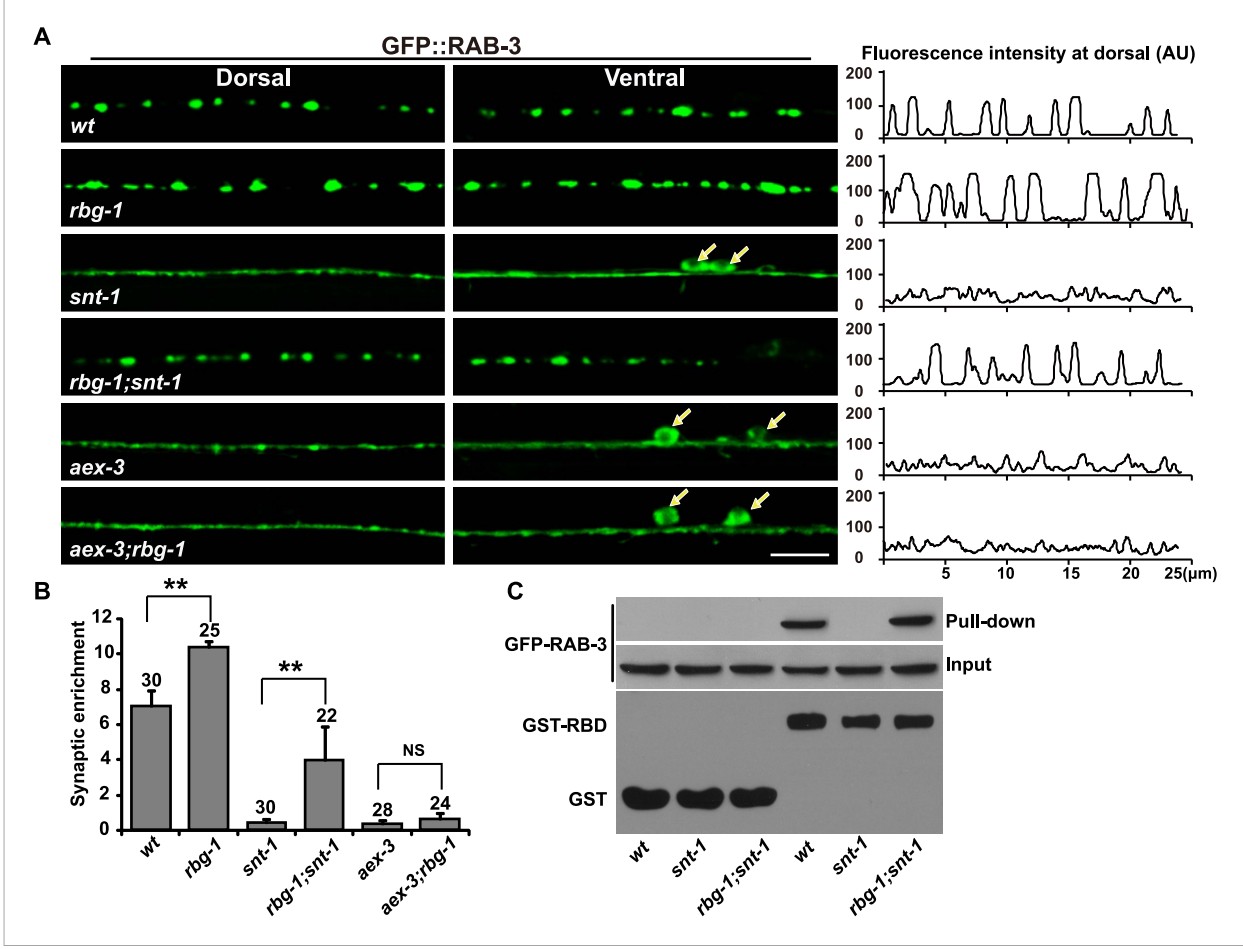

**Figure 4**. RAB-3 GAP mutations suppress the *snt-1* mutant phenotype. (**A**) The punctate distribution of GFP::RAB-3 is restored in *rbg-1;snt-1* animals, while the *aex-3* phenotype could not be suppressed by mutation of *rbg-1*. Yellow arrows indicate the cell bodies along the ventral cord. Scale bar, 5 µm. A representative line-scanning image for each genotype is shown in the right panel. (**B**) Quantification of the synaptic enrichment of GFP::RAB-3 signal in the genotypes shown in (**A**). Data are represented as mean ± SD. **p < 0.01; NS, not significant. (**C**) The amount of GTP-RAB-3 pulled down by RBD is increased in *rbg-1;snt-1* animals compared to *snt-1*.

The following figure supplement is available for figure 4:

**Figure supplement 1**. *rbg-2* suppresses the *snt-1* mutant phenotype.

activity is indeed increased in *snt-1* mutants, we would expect that loss of GAP function will suppress the *snt-1* mutant phenotype. Indeed, in *rbg-1;snt-1* double mutants, we found that the diffuse GFP::RAB-3 phenotype of *snt-1* single mutants is significantly suppressed (**Figure 4A,B**). Furthermore, mutation of the *rbg-2* gene also suppressed the diffuse RAB-3 phenotype in *snt-1* mutants (**Figure 4—figure supplement 1E**). In contrast, the diffuse GFP::RAB-3 signal caused by *aex-3* mutation could not be suppressed by *rbg-1* (**Figure 4A,B**). We next performed RIM2-RBD pull-down assays to test whether the GTP-RAB-3 level was restored in *rbg-1;snt-1* mutants. In contrast to the greatly reduced GTP-RAB-3 level in *snt-1* lysates, the amount of GTP-bound RAB-3 is significantly increased in *rbg-1;snt-1* samples (**Figure 4C**). Taken together, these results suggest that *snt-1* indeed regulate the RAB-3/SV association specifically by inhibiting RAB-3 GTP hydrolysis.

## Rab3GAP1/RBG-1 localizes on SVs

As the catalytic subunit of Rab3 GAP, RBG-1 can interact with RAB-3 (**Figure 4—figure supplement 1F**). To understand how SNT-1 inhibits RAB-3 GTP hydrolysis, we examined the sub-cellular localization of RBG-1. We created a functional mCherry::RBG-1 construct and injected it into *rbg-1*

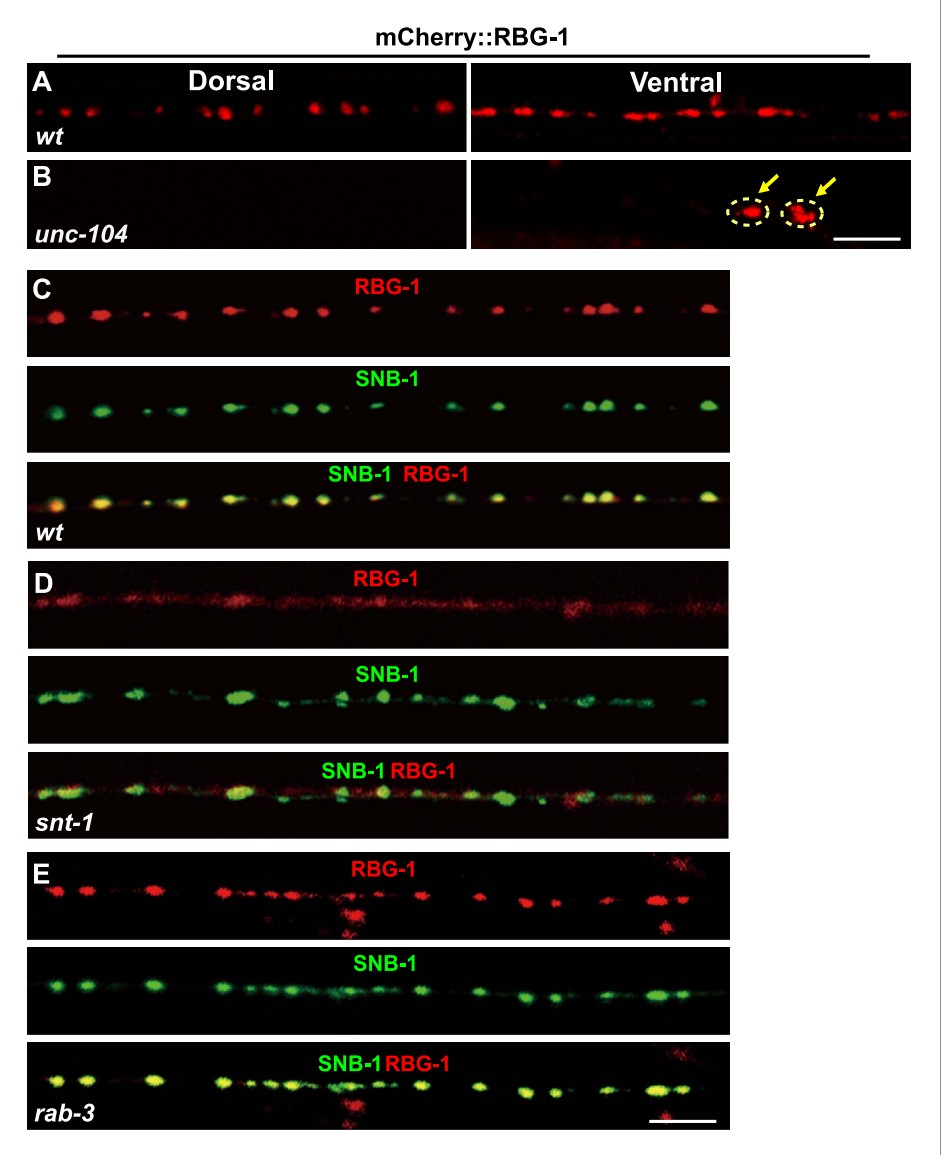

**Figure 5**. Localization of RBG-1 on synaptic vesicles requires SNT-1. (**A** and **B**) mCherry::RBG-1 has a punctate distribution in wild type (**A**) but accumulates in cell bodies (yellow arrows) in *unc-104* mutants (**B**). (**C**) mCherry::RBG-1 (red) is co-localized with SNB-1::GFP puncta (green). (**D**) In *snt-1* mutants, mCherry::RBG-1 loses its punctate localization and becomes diffuse in axons, while SNB-1::GFP retains its punctate pattern. (**E**) mCherry::RBG-1 retains its punctate distribution and is co-localized with SNB-1::GFP in *rab-3* mutants. Scale bars, 5 µm.

mutant animals. In wild-type animals, the mCherry::RBG-1 signal displayed a punctate expression pattern along the ventral and dorsal cords. Double staining further showed that the mCherry::RBG-1 puncta co-localized with the SV marker SNB-1::GFP (*Figure 5C*). Next, we tested whether the punctate localization of RBG-1 relies on the UNC-104-based intracellular transport system like other SV-associated proteins. We found that in *unc-104* mutants, the mCherry::RBG-1 puncta no longer appeared in the putative synaptic region; instead they were retained in the cell bodies (*Figure 5B*). Together, the data above suggest that RBG-1 is localized on SVs.

## SNT-1 is required for the SV localization of RBG-1

SNT-1 resides on SVs and its function in RAB-3 localization is executed by RAB-3 GAP. Could the SV localization of RBG-1 be regulated by *snt-1*? We examined RBG-1 distribution in *snt-1* animals and found that the RBG-1/SV co-localization is lost and mCherry::RBG-1 fluorescence becomes diffuse

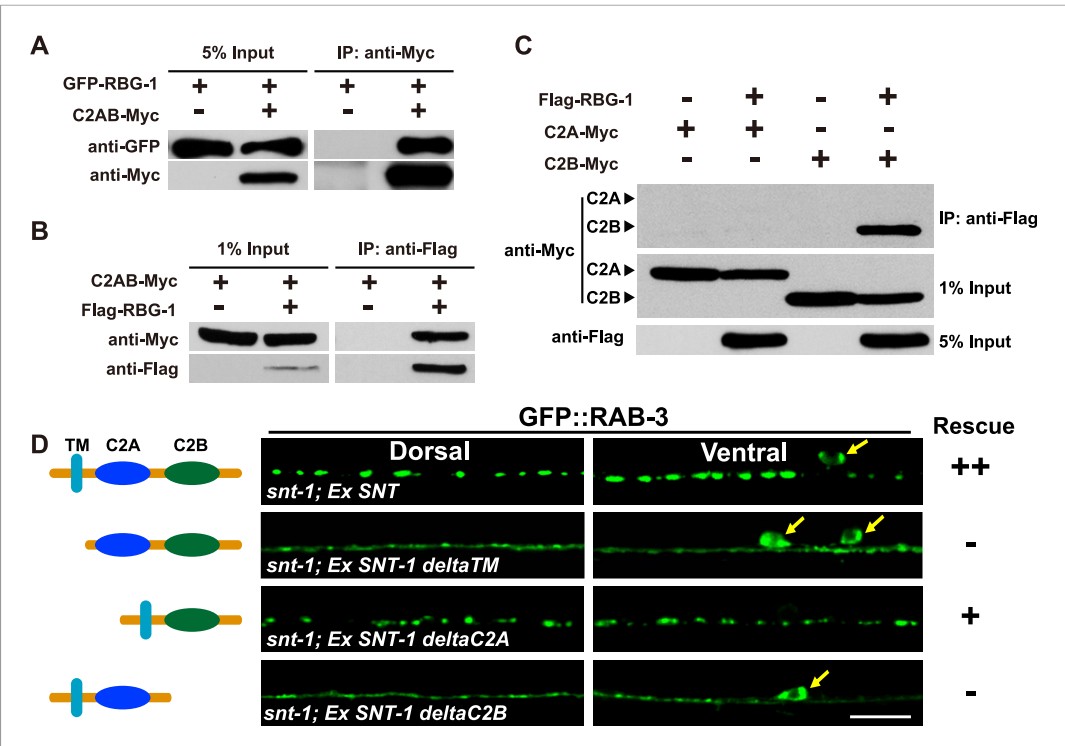

**Figure 6**. RBG-1 associates with the C2B domain of SNT-1. (**A**) RBG-1 is precipitated by the intracellular domain (C2AB) of SNT-1. (**B**) The SNT-1 intracellular domain is precipitated by RBG-1. (**C**) The C2B domain of SNT-1 binds to RBG-1. (**D**) SNT-1 without the C2B domain fails to rescue the *snt-1* mutant phenotype. Yellow arrows indicate cell bodies. The schematic diagram shows the transmembrane (TM) and intracellular calcium-binding domains (C2A and C2B) of SNT-1. Scale bar, 5 μm.

(*Figure 5D*). In contrast, SNB-1::GFP still retains its punctate expression pattern in *snt-1* animals, similar to wild type (*Figure 5D*). Therefore, SNT-1 is required for the SV localization of RBG-1.

RBG-1 binds to its substrate RAB-3 (*Figure 4—figure supplement 1E*). Thus, we also tested whether the SV localization of RBG-1 is controlled by RAB-3. In *rab-3(js49)* mutants, we found that mCherry::RBG-1 retains its punctate expression and still co-localizes with SNB-1::GFP (*Figure 5E*), suggesting that the SV localization of RBG-1 does not rely on RAB-3 protein.

## Rab3GAP1/RBG-1 binds to the C2B domain of SNT-1

The SNT-1-dependent SV association of RBG-1 suggests a direct association between SNT-1 and RBG-1. We co-expressed full-length RBG-1 and the cytosolic domain of SNT-1 (C2AB) in HEK293FT cells. After affinity purification, the RBG-1 protein was incubated with the SNT-1 C2AB fragment. In contrast to the mock-transfected sample, RBG-1 was effectively co-precipitated with the cytosolic region of SNT-1 (*Figure 6A*). The SNT-1 cytosolic portion was also co-precipitated by RBG-1 (*Figure 6B*). We next asked which domain of the SNT-1 cytosolic region is required for this binding. The cytosolic region of SNT-1 contains a C2A and a C2B motif. When the C2A domain was deleted, the remaining C2B motif retained the RBG-1 binding activity (*Figure 6C*). In contrast, when C2B was removed, the C2A domain alone could not bind to RBG-1 (*Figure 6C*). Therefore, the C2B domain is required for SNT-1 binding to RBG-1.

We further investigated whether the function of SNT-1 in regulating the RAB-3/SV association is mediated through the C2B domain. In comparison with full-length SNT-1, which fully rescues the GFP::RAB-3 mis-localization defect in *snt-1* mutants, we found that SNT-1 without the C2B domain (deltaC2B) had no rescue effect (*Figure 6D*). In contrast, when the C2A domain is removed (deltaC2A) from SNT-1, the remaining protein still possesses the rescue activity (*Figure 6D*). Together, these data suggest that the C2B domain is critical for SNT-1 function. SNT-1 lacking the trans-membrane domain (deltaTM) also failed to rescue the diffuse RAB-3 phenotype (*Figure 6D*), suggesting that the SV

localization function of SNT-1 is required in addition to the C2B domain for regulating RAB-3/SV association in vivo.

## Ca²⁺ treatment decreases the binding between SNT-1 and Rab3GAP1/RBG-1

Ca²⁺-mediated exocytosis activates the dissociation of Rab3 from the SV membrane (*Fischer von Mollard et al., 1991*; *Fischer von Mollard et al., 1994*). Could synaptotagmin 1, as the Ca²⁺ sensor for SV exocytosis, be the trigger to initiate the Rab3 SV dissociation process? We showed above that SNT-1 directly associates with Rab3GAP1/RBG-1. Therefore, we wondered whether the Ca²⁺ level could affect the binding of SNT-1 to RabGAP1/RBG-1, and whether the inhibition of RAB-3 GAP by SNT-1 is relieved by Ca²⁺ binding, thus allowing dissociation of RAB-3 from the SV during exocytosis. To test the ideas above, we asked whether the presence of Ca²⁺ disrupts the binding between SNT-1 and RBG-1. We purified RBG-1 and SNT-1 proteins and performed co-IP experiments with increasing concentrations of Ca²⁺. A Ca²⁺ concentration of 0.5 mM or 1 mM significantly compromised the RBG-1/SNT-1interaction (*Figure 7A,B*). Thus, upon Ca²⁺ binding, SNT-1 releases RBG-1.

Next, we asked whether SNT-1-RBG-1 binding is still affected by Ca²⁺ treatment if the Ca²⁺-binding sites are removed from SNT-1. The amino acids critical for Ca²⁺ binding in the C2A domain (D248 and D250) and the C2B domain (D383 and D385) were mutated, and the resulting SNT-1 mutant protein (C2A*B*) was purified and tested for its ability to bind RBG-1. We found that the RBG-1-binding capability of C2A*B* was high in the absence or presence of Ca²⁺ (*Figure 7C,D*), suggesting that the Ca²⁺-binding activity of SNT-1 is essential for attenuation of the SNT-1/RBG-1 interaction when Ca²⁺ is present.

## Dissociation of RAB-3 from SVs relies on the Ca²⁺-binding activity of SNT-1

Ca²⁺ treatment decreases the binding between SNT-1 and RBG-1. Is the inhibition of RAB-3 GAP by SNT-1 alleviated when the Ca²⁺concentration rises? If so, SNT-1 that lacks Ca²⁺-binding capability will fail to release RAB-3 GAP during Ca²⁺ influx, and thus the dissociation of RAB-3 from SVs will be inhibited. To test this hypothesis, we analyzed the GFP::RAB-3 pattern in transgenic animals that express different mutant forms of SNT-1. The two amino acids necessary for Ca²⁺ binding in the C2A domain of SNT-1 are D248 and D250. We created the SNT-1 mutant C2A* by replacing these two aspartic acids with alanines and found that C2A* could rescue the GFP::RAB-3 mis-localization defect (*Figure 7E*). In addition to this rescue phenomenon, we noticed that the GFP::RAB-3 puncta were enlarged and the GFP signal was more enriched in the punctate regions in comparison to wild-type animals over-expressing *snt-1* (*Figure 7E,F*). A similar GFP::RAB-3 enrichment effect was observed in worms expressing the C2B* mutant form of SNT-1, which contains a C2B domain that cannot bind Ca²⁺ (D383 and D385 were replaced with alanines) (*Figure 7E,F*). We further replaced the Ca²⁺-binding sites in both C2A and C2B (C2A*B*). When this construct was expressed, the GFP::RAB-3 signal was also increased in the punctate regions (*Figure 7E,F*).

Could the enhanced GFP:RAB-3 signal in the puncta indeed reflect the failure of exocytosis? Previous studies suggest that RAB3 dissociation from SV membranes is inhibited when SV exocytosis is disrupted by blocking either Ca²⁺ influx or membrane fusion (*Fischer von Mollard et al., 1991*; *Fischer von Mollard et al., 1994*; *Stahl et al., 1996*). We examined two exocytosis mutants, *unc-2* and *unc-13*. *unc-2* encodes the alpha subunit of the voltage-gated Ca²⁺ channel (*Schafer and Kenyon, 1995*). In *unc-2* mutants, we found that the GFP signal is significantly enriched within the punctate regions and the GFP::RAB-3 puncta are larger and brighter compared to wild type (*Figure 7—figure supplement 1A,B*). *unc-13* encodes the Munc13 homolog in worms and loss of function of *unc-13* results in blockage of membrane fusion during SV exocytosis (*Aravamudan et al., 1999*; *Richmond et al., 1999*; *Varoqueaux et al., 2002*). We found that the GFP signal was enhanced in the punctate regions in *unc-13* mutants, as in *unc-2* mutants (*Figure 7—figure supplement 1A,B*). Blocking SV exocytosis resulted in failure of RAB-3 to dissociate from SVs; thus the enlarged GFP::RAB-3 puncta indeed indicate the decreased dissociation of RAB-3 protein from SVs. Taken together, the data above indicate that the Ca²⁺-binding capability is essential for SNT-1 function in mediating the RAB-3 SV dissociation induced by Ca²⁺-triggered exocytosis.

Interestingly, in *unc-13;snt-1* or *unc-2;snt-1* double mutants, the RAB-3::GFP signal is diffuse throughout whole neuronal cells (*Figure 7—figure supplement 1A,B*), as seen in *snt-1* single mutants.

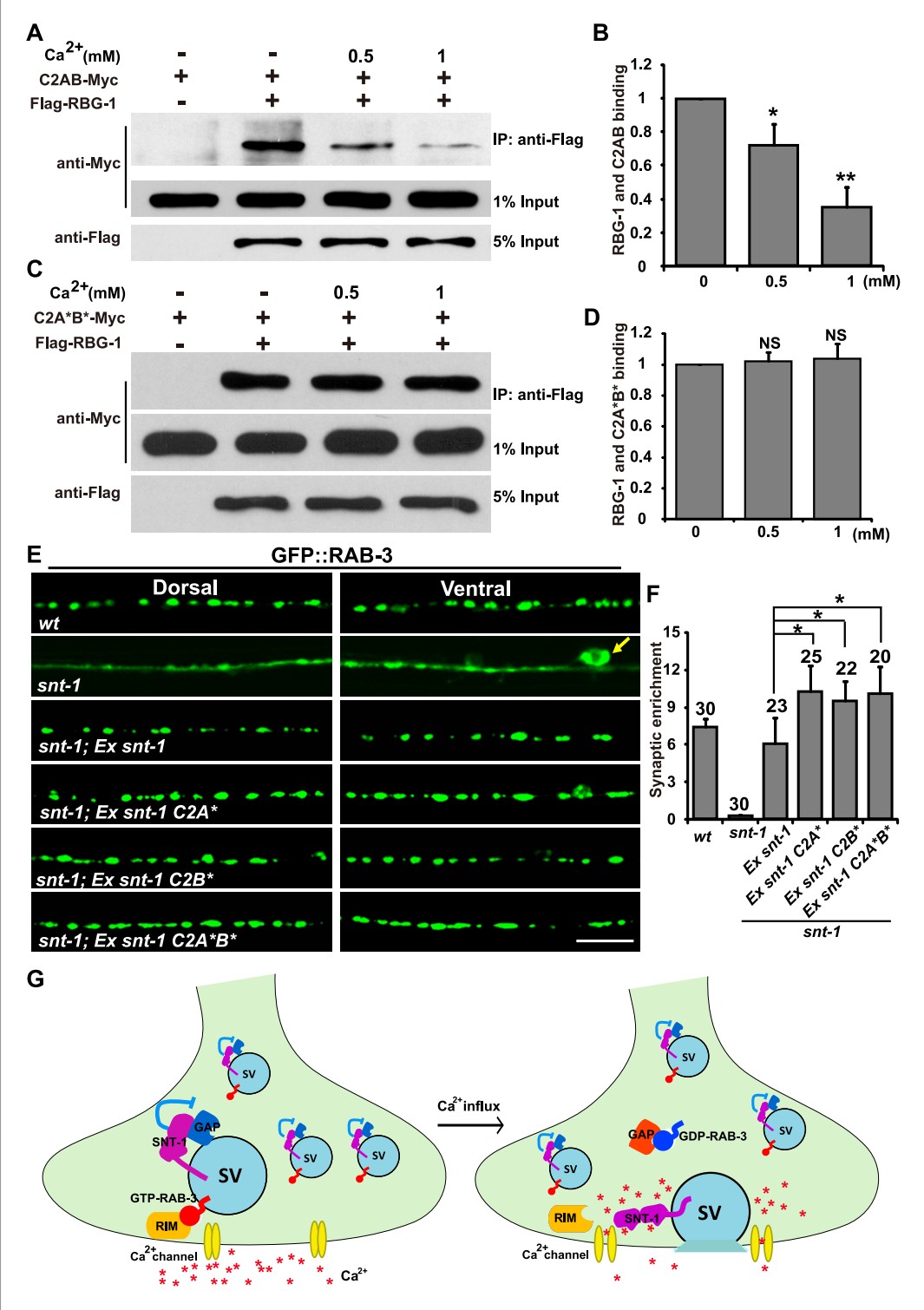

**Figure 7**. Dissociation of RAB-3 from synaptic vesicles requires the $Ca^{2+}$-binding activity of SNT-1. (**A**) $Ca^{2+}$ treatment diminishes the binding between RBG-1 and the intracellular domain (C2AB) of SNT-1. (**B**) Quantification of the relative binding between RBG-1 and the C2AB domain upon $Ca^{2+}$ treatment. (**C**) SNT-1 without $Ca^{2+}$-binding sites (C2A\*B\*) still binds to RBG-1 in the presence of $Ca^{2+}$. (**D**) Quantification of the relative binding between RBG-1 and the C2A\*B\* domain upon $Ca^{2+}$ treatment. (**E**) Mutant SNT-1 proteins without $Ca^{2+}$-binding activity stabilize RAB-3 on SVs. *Ex snt-1*, over-expression of SNT-1; *Ex snt-1 C2A\** and *Ex snt-1 C2B\**, over-expression of SNT-1 with mutant C2A domain or C2B domain, respectively; *Ex snt-1 C2A\*B\**, over-expression of SNT-1 with mutant C2A and C2B

*Figure 7. Continued*

domains. Scale bar, 5 µm. (**F**) Quantification of the synaptic enrichment in the genotypes shown in (**E**). (**G**) SNT-1 functions as a molecular switch controlling RAB-3/SV association and disassociation during SV exocytosis. Data are represented as mean ± SD. *p < 0.05; **p < 0.01; NS, not significant.

The following figure supplement is available for figure 7:

**Figure supplement 1**. Exocytosis is uncoupled from RAB-3 synaptic vesicle dissociation in *snt-1* mutants.

Because the diffuse RAB-3::GFP signal suggests a dissociation of RAB-3 from SV membranes, the above observation implies that in the absence of SNT-1, the dissociation of RAB-3 from SVs can occur even when $Ca^{2+}$-induced exocytosis is blocked. As a consequence, the coupling between SV exocytosis and RAB-3 dissociation is disrupted when SNT-1 is missing. These data are consistent with the inhibition-of-inhibition role of SNT-1 on RAB-3/SV association. Together, these results suggest that SNT-1 plays a dual role in the RAB-3/SV cycle, inhibiting dissociation of RAB-3 from SVs during the resting state (no $Ca^{2+}$ influx) and triggering dissociation of RAB-3 from SVs upon $Ca^{2+}$ binding (*Figure 7G*).

## Discussion

Acute and precise neuronal activity requires precise coordination between the SV cycle and the Rab3 cycle. As the trigger of regulated vesicle secretion, synaptotagmin 1 is known to bind the membrane and the SNARE complex to give the final push for complete assembly of the SNARE complex for membrane fusion (*Davletov and Südhof, 1993*; *Chapman and Davis, 1998*; *Dai et al., 2007*; *Choi et al., 2010*; *Vrljic et al., 2010*). Here, we revealed that SNT-1/synaptotagmin 1 functions as an on-and-off switch to regulate Rab3 membrane association, thus facilitating repeated release. Our study has many important implications.

In the resting state, Rab3 protein is associated with SVs in the GTP-bound form, and Rab3 GTP, together with its corresponding effectors, docks the vesicles at the active zone region (*Sudhof, 2004*). However, Rab3 GAP, the negative regulator of Rab3, is also enriched in the synaptic fraction and is localized on SVs (this study) (*Oishi et al., 1998*). How therefore is the active form of Rab3, and hence the proper docking complex, maintained when GAP is close by? Based upon our study, it is entirely possible that synaptotagmin 1 prevents the hydrolysis of Rab3 GTP by directly sequestering or inhibiting Rab3 GAP before $Ca^{2+}$ influx. In fact, the SV localization of Rab3GAP1/RBG-1 is particularly interesting, given the fact that the Rab3 cycle must be spatially regulated so that Rab3 is kept in close proximity to SVs for repeated neurotransmitter release. Indeed, when synaptotagmin 1/SNT-1 binds $Ca^{2+}$, the inhibition on RAB-3 GAP is alleviated, so the locally enriched Rab3 GAP can freely and quickly access GTP-Rab3 and hydrolyze Rab3 GTP to GDP. Together with the functional $Ca^{2+}$switch, synaptotagmin 1 can therefore efficiently coordinate the Rab3 cycle with the SV cycle. The concentration of $Ca^{2+}$ needed for exocytosis ranges from ∼10 to 200 µM in the nerve terminals, so the $Ca^{2+}$ concentration that is required to release RBG-1 from SNT-1 is relatively high (500 µM). However, it is known that the rather low affinity of synaptotagmin 1 for $Ca^{2+}$ (*Ubach et al., 1998*; *Fernandez et al., 2001*) can be strongly affected by the presence of synaptotagmin-binding partners, especially membrane lipids (*Chapman, 2002*). Therefore, it will be interesting to test whether plasma membrane-enriched lipids such as PIP2 (*Lee et al., 2010*) can act synergistically with $Ca^{2+}$ to regulate the binding of SNT-1 to RBG-1.

At the molecular level, synaptotagmin 1 is known to bind the plasma membrane and the SNARE complex in the presence of $Ca^{2+}$ (*Davletov and Südhof, 1993*; *Chapman and Davis, 1998*; *Fernandez et al., 2001*; *Dai et al., 2007*; *Choi et al., 2010*; *Vrljic et al., 2010*). The interaction of synaptotagmin with membranes and SNARE proteins has well-documented consequences, including creating local positive membrane curvature and displacing the clamping factor complexin from the SNARE complex (*Martens et al., 2007*; *Hui et al., 2009*). Combining these previous reports with data presented in this study, we think that synaptotagmin-$Ca^{2+}$ may not only give the final push for complete assembly of the SNARE complex for membrane fusion, but also play a role in terminating vesicle docking by indirectly deactivating Rab3, thus facilitating repetitive transmitter release.

Among more than 60 Rabs in humans and mice, Rab3 and Rab27 seem to be specifically involved in stimulated secretion in a variety of secretory cells (*Fukuda, 2008*, *2013*). Synaptotagmins have also evolved specifically to regulate secretion. Interestingly, the regulatory role of SNT-1 on RAB-3 does

not extend to RAB-27, which is functionally closely related to RAB-3 (*Mahoney et al., 2006*). Simultaneous knockdown of Rab3 and Rab27 causes secretion defects more severe than single knockdown in worms and PC12 cells (*Mahoney et al., 2006*; *Tsuboi and Fukuda, 2006*). Both Rab27 and Rab3 are localized on SVs and can bind to RIM, thereby docking the vesicles (*Wang et al., 1997*; *Fukuda, 2003*). Rab3 and Rab27 share the same GEF, which is AEX-3 in worms and DENN/MADD in mammals (*Levivier et al., 2001*; *Coppola et al., 2002*; *Mahoney et al., 2006*). In contrast, Rab3 GAP serves as a specific GAP for Rab3 (*Fukui et al., 1997*; *Nagano et al., 1998*; *Itoh and Fukuda, 2006*). Here, SNT-1 apparently only affects the RAB-3 cycle and this action is mediated by inhibition of the Rab3-specific GAP. Therefore, although Rab3 and Rab27 play redundant roles in SV exocytosis, they can be differentially controlled through their specific regulators. We would like to point out that humans and worms have multiple synaptotagmins, and it remains to be determined whether any of the other synaptotagmins play a similar regulatory role on Rab27.

Rab3 GAP consists of the catalytic subunit Rab3GAP1 and the noncatalytic subunit Rab3GAP2 (*Fukui et al., 1997*; *Nagano et al., 1998*). Rab3GAP1 and Rab3GAP2 form a complex in vitro and co-immunoprecipitate in vivo (*Nagano et al., 1998*). Although Rab3GAP2 does not affect the in vitro GAP activity of Rab GAP1, it may act to stabilize, regulate, or localize Rab3GAP1 correctly in cells. The functional characteristics are consistent with their close biochemical interactions. Loss-of-function mutations in Rab3GAP1 and Rab3GAP2 produce clinically almost indistinguishable conditions, Warburg Micro syndrome and Martsolf syndrome, characterized by brain, eye, and endocrine abnormalities (*Aligianis et al., 2005*, *2006*). We have now revealed that both Rab3GAP1/*rbg-1* and Rab3GAP1/*rbg-2* mutations can suppress the RAB-3 mis-localization phenotype in *snt-1* mutants, implying that Rab3GAP does indeed function as a complex to participate in the SNT-1-mediated regulation of the RAB-3 cycle. However, we noticed that loss of function of *rbg-2* alone leads to additional synaptic or axonal defects (data not shown) compared to loss of *rab-3* or *rbg-1*. This suggests that RBG-2 may play roles in nervous system development other than forming the Rab3GAP complex with RBG-1 during SV exocytosis.

The GTP-bound active form of Rab promotes membrane trafficking by interacting with specific effectors. In contrast with Rab effectors that function in secretory vesicle trafficking, relatively little is known about the specific Rab GEFs and GAPs and how they are regulated during vesicle secretion. Evidence that the sub-cellular localization and activity of RAB-3 GAP can be regulated by synaptotagmin/SNT-1 strongly hints that regulators of Rabs could be subjects for active manipulation during various types of intracellular membrane trafficking. A recent intriguing study is in agreement with our notion. In the amoeba *Dictyostelium discoideum*, vacuolar $Ca^{2+}$ release activates the Rab GAP CnrF, thus subsequently down-regulating Rab11a (*Donato et al., 2013*). Taken together, current data suggest a vital role of $Ca^{2+}$ as the functional switch for regulated secretion processes. Because the key principles and regulatory components of different intracellular vesicle trafficking events are broadly conserved, the mechanism that we have uncovered is likely to represent a conserved mode of action.

Understanding how the sequential activation of Rab GTPases is achieved during vesicle trafficking is a central theme of cell biology. This in turn raises the question of how regulatory Rab GEFs or Rab GAPs are activated in the right place and at the right time. Synaptotagmins act as the primary $Ca^{2+}$-sensors in most forms of $Ca^{2+}$-induced exocytosis, as exemplified by synaptic transmission. Our studies have proposed the elegant molecular machinery by which Rab GAP can be temporally and spatially regulated in response to acute cellular signals, so that correctly activated Rabs can perform their appropriate functions and allow vesicle fusion to occur in an orderly fashion.

## Materials and methods

### Worm strains and mutagenesis

Strain maintenance and genetic manipulations were performed as described (*Brenner, 1974*). Strains used in this study are: **LG I**: EG2710 [*unc-57(ok310)*], CB450 [*unc-13(e450)*], VC2481 [*rbg-2(ok3195)*], CB47 [*unc-11(e47)*]. **LGII**: NM204 [*snt-1(md290)*], CB1265 [*unc-104(e1265)*], MT6977 [*snt-1(n2665)*], RM1606 [*snt-1(md172)*], RM1603 [*snt-1(md125)*], RM1620 [*snt-1(md220)*], NM791 [*rab-3(js49)*]. **LGIII**: XD1366 [*rep-1(xd56)*], NM1278 [*rbf-1(js232)*]. **LGIV**: EG3027 [*unc-26(s1710)*], CB169 [*unc-31(e169)*]. **LGV**: XD1925 [*hmgs-1(xd145)*], CB268 [*unc-41 (e268)*], NM467 [*snb-1(md247)*], RM956 [*ric-4 (md1088)*]. **LGX**: XD1199 [*aex-3(xd58)*], RB1453 [*rbg-1(ok1660)*], CB81 [*unc-18(e81)*], CB55 [*unc-2 (e55)*], CB102 [*unc-10(e102)*], CX51 [*dyn-1(ky51)*]. Additional strains are: XD2188 [*xdEx1380*; P*aex-3*::

AEX-3::GFP], XD2702 [xdEx1214; P*unc-25*::mCHERRY::RBG-1], XD3132 [xdEx1461; P*hmr-1b*::SNT-1FL], XD3017 [xdEx1419; P*hmr-1b*::SNT-1 deltaTM], XD3134 [xdEx1463; P*hmr-1b*::SNT-1 deltaC2A], XD3034 [xdEx1398; P*hmr-1b*::SNT-1 deltaC2B], XD3032 [xdEx1396; P*hmr-1b*::SNT-1 C2A*], XD3033 [xdEx1397; P*hmr-1b*::SNT-1C2B*], XD3133 [xdEx1462; P*hmr-1b*::SNT-1C2A*B*]. Mutagenesis was carried out in the xdIs7 (P*hmr-1b*::GFP::RAB-3) strain treated with ethylmethane sulfonate. From 5000 genomes, 13 mutations were isolated. Subsequent genetic and molecular analysis revealed that we had isolated four alleles of rep-1(xd56, xd138, xd139, and xd142), six alleles of aex-3(xd58, xd60, xd137, xd143, xd148, and xd149), and three alleles of hmgs-1(xd128, xd129, and xd145). rep-1 (xd138), rep-1(xd139), hmgs-1(xd128) and hmgs-1(xd129) animals are larval lethal. The rest of the identified mutants are fertile.

## DNA constructs and transgenic animals

Promoters, GFP, mCherry, and various cDNA or genomic DNA fragments were cloned into the deltapSM vector through standard procedures. Site-directed mutagenesis was performed using standard PCR-based methods. Transgenic animals were produced as previously described (Song et al., 2010). Integrated strains were obtained by UV irradiation. All integrated transgenic animals were out-crossed at least 3 times.

## GTP-RAB-3 pull-down assay

DNA fragments containing the RBD domains of rat RIM2 were inserted into the pGEX-4T-3 vector. Expression and purification of GST fusion proteins in *E. coli* were carried out according to standard procedures. Worms with different genotypes were collected and washed in M9 buffer. 800 μl of homogenizing buffer (50 mM Tris-Cl pH8.0, NaCl 150 mM, 0.5% sodium deoxycholate, 1% Triton-X 100) was added and samples were disrupted with a Dounce homogenizer (Cheng-He Company, Zhuhai, China) (Chen et al., 2010) on ice for 5 min. Debris was removed by centrifuging at 12,000 rpm for 10 min at 4°C. The amount of GFP-RAB-3 input in each experiment was equalized before the pull-down assay. The worm lysates were incubated for 4 hr at 4°C with GST-tagged RBD RIM2 coupled to glutathione-Sepharose 4B (GE Healthcare, USA). After washing three times, the GFP-RAB-3 level was analyzed by 10% SDS-PAGE followed by standard western blotting with an anti-GFP antibody (1:5000 dilution) (Santa Cruz Biotechnology).

## Co-immunoprecipitation

To express proteins in HEK293FT cells, cDNA fragments were amplified and cloned into modified pcDNA™3.1/myc-HIS(−) or pFLAG-CMV-2 vectors through standard procedures. HET293FT cells were cultured in DMEM medium supplemented with 12% FBS. Plasmid transfections were carried out using Lipofectamine 2000 (Invitrogen, USA). 24 hours after transfection, cells were harvested and lysed for 10 min at 4°C. After centrifugation, the supernatants were incubated with anti-FLAG or anti-myc beads at 4°C for 4 hr. Samples were resolved by standard immunoblotting techniques. For co-immunoprecipitation experiments with purified proteins, the immunoprecipitated samples were eluted with elution buffer (Thermal Scientific, USA) and neutralized with Tris buffer (pH 9.0). For GFP-tagged proteins, anti-GFP antibody (Abcam, USA) was incubated with the protein supernatant.

## Cell fractionation

Worms with different genotypes were collected and washed in M9 buffer. 500 μl of lysis buffer (250 mM sucrose, 50 mM Tris–HCl with pH6.8, 1 mM EDTA) were added and worm samples were homogenized with a Dounce homogenizer (Cheng-He Company, Zhuhai, China) (Chen et al., 2010) on ice for 15 min. The nuclear pellet was removed by centrifuging at 3000 rpm for 10 min at 4°C. The supernatant was further centrifuged at 40,000 rpm for 1 hr. The new supernatant was collected as the cytosolic fraction. The pellet was further washed and centrifuged at 40,000 rpm for 45 min. All samples were mixed with 2xSDS loading buffer before 10% SDS-PAGE gel analysis. The GFP-RAB-3 and tubulin levels in each fraction were analyzed by standard western blotting procedures.

## Microscopy and image analysis

Images were captured using a Plan-Apochromat 40X/1.4 objective on an Olympus confocal microscope. All images were taken at the young adult stage unless specifically indicated. Images were analyzed with custom Image J software. Two main parameters were determined: puncta number (PN) and synaptic

enrichment (SE) (*Ch'ng et al., 2008*). These were calculated from the punctal fluorescence (PF), which measures the signal intensity at the pre-synaptic specialization, and the inter-punctal fluorescence (IPF), which measures the signal intensity in axons between synapses. An individual punctum is defined when the peak PF/average peak IPF is ≥2. Synaptic enrichment is defined as total PF/total IPF within a 100-µm length of middle dorsal cord region. All data are shown as mean ± SD. Statistical analyses were performed with Student's t-test. For each genotype, more than 20 animals were imaged and analyzed.

## Acknowledgments

We thank Dr Eric Jorgensen, Dr James Rand, Dr Yishi Jin, Dr Xiaochen Wang, Dr Jianyuan Sun, Dr Yuji Kohara, the *C. elegans* Gene Knockout Consortium, the TransgeneOme Project, and the Caenorhabditis Genetics Center for providing reagents, strains, and technical support. We thank Dr Zhaohui Wang for help with confocal imaging. This work was supported by the National Natural Science Foundation of China (31130023, 31222026, and 31490593), and grants from the National Basic Research Program of China (Y121044691), and the Chinese Academy of Sciences (KSCX-EW-R-05).

## Additional information

### Funding

| Funder | Grant reference | Author |
| --- | --- | --- |
| National Natural Science Foundation of China (NSFC) | 31130023 | Mei Ding |
| Ministry of Science and Technology, Taiwan | National Basic Research Program of China Y121044691 | Mei Ding |
| Chinese Academy of Sciences (CAS) | KSCX-EW-R-o5 | Mei Ding |
| National Natural Science Foundation of China (NSFC) | 31222026 | Mei Ding |
| National Natural Science Foundation of China (NSFC) | 31490593 | Mei Ding |

The funders had no role in study design, data collection and interpretation, or the decision to submit the work for publication.

### Author contributions

YC, Conception and design, Acquisition of data, Analysis and interpretation of data; JW, Acquisition of data, Analysis and interpretation of data; YW, Acquisition of data, Contributed unpublished essential data or reagents; MD, Conception and design, Drafting or revising the article

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
