## [Decision Letter]

Thank you for sending your work entitled “Synaptotagmin 1 directs repetitive release by coupling vesicle exocytosis to Rab3 cycle” for consideration at *eLife*. Your article has been favorably evaluated by Randy Schekman (Senior editor) and 3 reviewers, one of whom is a member of our Board of Reviewing Editors.

The Reviewing editor and the other reviewers discussed their comments before we reached this decision, and the Reviewing editor has assembled the following comments to help you prepare a revised submission. In principle, all reviewers considered your work as interesting but there are several conclusions that need to be substantiated in a revision. First, the notion that SNT inhibits Rab3-GAP needs more support. One approach would be to examine RBD RAB-3 pulldowns in *snt-1*; *rbg-1* double mutants. This experiment is a better test for the hypothesis that SNT-1's effects on RAB-3/GTP levels are caused by a change in RBG-1 activity. Even better (if feasible) would be to carry out GAP assays in vitro.

Second, the reviewers questioned whether RAB-3 was cytosolic or plasma membrane associated in *snt-1* mutants. If the diffuse RAB-3 is on the plasma membrane, that would not support the model for increased GAP activity. If RAB-3 is cytosolic, that would provide independent support for the altered GTP state (i.e. confirming the RBD pull down). For these reasons the reviewers request direct fractionation experiments (analyzing Rab3 levels in the membrane and soluble fractions) to solidify this key conclusion.

Third, the sensitivity of the *snt-1* and *rab3* single mutants and the *snt-1/rab3* double mutant to the inhibitor aldicarb should be tested. If the authors are correct, there should be no increased sensitivity of the double mutant.

Fourth, add an *rbg-1* rescue experiment.

Finally, some of the referees were concerned about the nonphysiologically high concentrations of calcium that are required to see the effect. This should be at least commented on by the authors.

[Editors' note: further revisions were requested prior to acceptance, as described below.]

Thank you for resubmitting your work entitled “Synaptotagmin 1 directs repetitive release by coupling vesicle exocytosis to the Rab3 cycle” for further consideration at *eLife*. Your revised article has been favorably evaluated by Randy Schekman (Senior editor), a member of the Board of Reviewing Editors, and the two original referees. The manuscript has been improved but there was one minor issue raised by one of the referees that needs to be addressed before acceptance, as outlined below: “The methods state that worm extracts were prepared by dounce homogenization for the biochemistry experiments. I thought that worms are not well disrupted by dounce, and that one needs to use sonication, french press, or a microfluidizer. Could you check with the authors that the methods are correctly reported?”

---

## [Author Response]

*First, the notion that SNT inhibits Rab3-GAP needs more support. One approach would be to examine RBD RAB-3 pulldowns in* snt-1*;* rbg-1 *double mutants. This experiment is a better test for the hypothesis that SNT-1's effects on RAB-3/GTP levels are caused by a change in RBG-1 activity. Even better (if feasible) would be to carry out GAP assays* in vitro.

We have performed the RBD RAB-3 pulldown in *snt-1;rbg1* double mutants and the corresponding data has been included in Figure 4. Basically, we found that the GTP-bound RAB-3 is increased in the double mutant compared with *snt-1* single mutant animals, further supporting the idea that SNT-1 can protect GTP-RAB-3 by inhibiting RAB-3 GAP activity. We have tried the in vitro GAP assay in our laboratory for a year. However, the extremely low expression level of Rab3GAP1/RBG-1 protein in both bacteria and insect cell lines prevents us from carrying out any further analysis. Hence, the GAP assays are not currently feasible.

*Second, the reviewers questioned whether RAB-3 was cytosolic or plasma membrane associated in* snt-1 *mutants. If the diffuse RAB-3 is on the plasma membrane, that would not support the model for increased GAP activity. If RAB-3 is cytosolic, that would provide independent support for the altered GTP state (i.e. confirming the RBD pull down). For these reasons the reviewers request direct fractionation experiments (analyzing Rab3 levels in the membrane and soluble fractions) to solidify this key conclusion*.

We have performed the cell fractionation experiments and the results are shown in Figure 3. Basically, the level of Rab-3 in the cytosolic fraction is greatly increased in *snt-1* mutants compared to wild type, which is consistent with our conclusion that SNT-1 regulates RAB-3 localization by altering its GTP status.

*Third, the sensitivity of the* snt-1 *and* rab3 *single mutants and the* snt-1/rab3 *double mutant to the inhibitor aldicarb should be tested. If the authors are correct, there should be no increased sensitivity of the double mutant*.

The *snt-1* (II:0.12 +/− 0.000 cM) and *rab-3* (II:-0.96 +/− 0.001 cM) genes are in a close proximity on the same chromosome. Therefore, we used two approaches to obtain the *snt-1 rab-3* double mutant. Firstly, we tried to use CRISPR-Cas9 to mutate the *snt-1* gene in the *rab-3(js49)* background. After guidance RNA sequence design, cloning and injection, we obtained 4 heritable transgenic lines. However, PCR and sequence analysis showed that none of the transgenic lines was carrying the corresponding *snt-1* mutation. Second, we tried to get the *snt-1 rab-3* recombinant directly. After crossing *rab-3(js49)* males to *snt-1(md290)*, from the heterozygote progenies, we obtained a total of 300 *snt-1(md290)* homozygotes. Among the 300 animals, three were *rab-3(js49)* heterozygotes. We singled out 72 animals from these three plates. However, PCR and sequencing analysis could not identify any *rab-3(js49)* homozygote animals, suggesting that the *snt-1 rab-3* double animals are probably lethal. Indeed, after monitoring the progeny from a single *snt-1 rab-3/snt-1 +* animal, we found that a quarter of the progeny were arrested at the later larval stage. Thus, the lethality of *snt-1 rab-3* doubles prevents us from doing the aldicarb assay.

*Fourth, add an* rbg-1 *rescue experiment*.

We have now included the *rbg-1* rescue data in Figure 4—figure supplement 1.

*Finally, some of the referees were concerned about the nonphysiologically high concentrations of calcium that are required to see the effect. This should be at least commented on by the authors*.

We have now commented on this point in the Discussion.

[Editors' note: further revisions were requested prior to acceptance, as described below.]

*The manuscript has been improved but there was one minor issue raised by one of the referees that needs to be addressed before acceptance, as outlined below:* “*The methods state that worm extracts were prepared by dounce homogenization for the biochemistry experiments. I thought that worms are not well disrupted by dounce, and that one needs to use sonication*, *french press, or a microfluidizer. Could you check with the authors that the methods are correctly reported?*”

We indeed used the Dounce homogenizer to break worms. This method was used by other researchers in worm community as well. Worms can be easily and well disrupted by this method. For instance, after homogenization 10 min on ice, protein concentration 10 µg/µl was obtained from worm sample composed of ∼100 µl worms and 300 µl buffer, which is comparable to other methods including sonication, French press, or microfluidizer. We now included the name of the company, which sells the homogenizers and the citation in the current manuscript in the subsection headed “Cell fractionation”.